# Development and use of integrated wetland condition index for lacustrine fringe wetlands of Lake Tana, Northwest Ethiopia

Yirga Kebede Wondim*, Ayalew Wondie Melese

Department of Aquatic and Wetland Ecosystem Management, School of Fisheries and Wildlife, College of Agriculture and Environmental Sciences, Bahir Dar University, Bahir Dar, Ethiopia

* yirgukeba@gmail.com

## Abstract

This research aimed to develop an integrated wetland condition index (IWCI) for lacustrine fringe wetlands (LFWs) in Lake Tana, Northwest Ethiopia. These wetlands have been highly impacted by recessionary agriculture, water hyacinth infestation, and both short- and long-term water level fluctuations (WLFs), as well as a heavy sediment load. The lacustrine wetland condition index was developed based on four key characteristics (hydrology, water quality, sediment quality, and wetland biota) that define wetlands. Value obtained from field and laboratory measurements of each wetland hydrology, water chemistry, and sediment characteristics indicator was normalized by allocating scores based on literature information and the results of this research project from three different disturbance-level lacustrine wetlands in Lake Tana. Macrophyte, diatom, and zooplankton indicator species tests (IndVal% and p (raw)) were calculated using PAST 4.14 (PAleontological STatistics). Scores were allocated on a scale from 1 to 10, in such a way that the highest scores reflected the best conditions and the lowest scores the most disturbed and unhealthy conditions. Within each of the four sub-indexes, multiple measures were equally weighted, while the four-wetland components (hydrology, water chemistry, sediment, and biota) were weighted according to their contribution to overall wetland condition. Weights were assigned using the Decision Support System Software of DEFINITE. Finally, the total IWCI was classified into five wetland condition categories, namely: very poor or very far from the reference (0–3), poor (3–5), moderate (5–7), good (7–9), and excellent wetland condition or reference condition (9–10). The results of IWCI of the low water level period indicated that eight wetland-sampling sites were found within a moderately impacted condition (44.44%); similarly, eight wetland-sampling sites were found within a good or mildly impacted condition; and only two wetland- sampling sites (11.11%) were found in an excellent or reference condition. Thus, this IWCI developed and tested in this study could be a tool to inform decision-makers on lacustrine wetland conservation and restoration priorities throughout the Lake Tana ecosystem.

**Data availability statement:** All relevant data are within the manuscript and its Supporting Information files.

**Funding:** The author(s) received no specific funding for this work.

**Competing interests:** The authors declare that they have no known competing financial interests or personal relationships that could have appeared to influence the work reported in this paper.

For managing and restoring shore area wetlands and making policy, lacustrine fringe wetlands monitoring sites need to be established and monitored for their condition using IWCI developed and tested by this study.

## Introduction

Lacustrine fringe wetlands are important ecotones for biodiversity conservation, spawning and refugium habitats, high primary productivity, and significant material exchange between aquatic and terrestrial ecosystems. Despite their crucial role as ecotones, lacustrine wetlands are highly vulnerable to anthropogenic activities, particularly water level fluctuations [1]. The assessment and monitoring of lacustrine fringe wetlands condition is an important component in the wise use of wetland resources. A useful and effective wetland condition index should incorporate indications of wetland function in addition to their structural characteristics. For the Index of Wetland Condition (IWC), "the state of the biological, physical, and chemical components of the wetland ecosystem and their interactions" is the definition of wetland condition [2].

The holistic concept of wetland condition is challenging to measure because it encompasses different issues such as ecosystem health, biodiversity, stability, sustainability, naturalness, wildness, and beauty [3]. Consequently, various wetland condition indices have been developed by different researchers, each focusing on specific aspects of wetland condition, such as the Index of Biological Integrity [4], Floristic Assessment Index [5], Landscape Development Index [6], Index of wetland condition from ecological variables used in hydrogeomorphic (HGM) assessment [7], Wetland Zooplankton Index [8], Performance of Phytoplankton Index [9], and Water Level Drawdown Index for aquatic macrophytes [10]. These wetland condition indexes that were developed separately by considering single wetland component could not provide sufficient information about the wetland condition.

The hydrogeomorphic (HGM) method to functional evaluation [11] and the Index of Biotic Integrity (IBI) approach to bioassessment of wetland conditions are recent attempts to develop guidelines for wetland assessment [12]. The HGM approach was specifically designed for developing rapid wetland function assessments, while the IBI, a multimetric bioassessment approach, is based on changes in the species composition of assemblages and the abundance of organisms, mainly macroinvertebrates. Both approaches focused on a wetland scale. Additionally, landscape ecology has shown how crucial it is to take landscape context into account in addition to local site characteristics when describing local ecological processes and ecological integrity [13]. It might also be helpful to apply landscape indicators that measure the spatial pattern of nearby wetland areas [14].

Few studies have focused on integrated wetland condition indices. In order to manage and conserve wetlands, there is a need for integrated methods of assessment [13]. Recently, efforts have been made to combine and alter a few indices to develop an integrative expression that considers different environmental components. For environmental assessment of wetlands, [15] integrated hydrogeomorphic

and Index of Biotic Integrity approaches into a single, more flexible, and broadly based approach. For Victoria, Australia, [16] developed a wetland condition index by integrating catchments, physical form, soils, water chemistry, hydrology, and biota for Victoria, Australia. The Landscape Development Intensity Index (LDI) and the Water Environment Index (WEI) were combined by [17] to create the Wetland Condition Index (WCI) for China's humid regions. [12] created the Florida Wetland Condition Index for Florida's isolated depressional wooded wetlands. [18] developed standardized measures of coastal wetland conditions for the Laurentian Great Lakes Basin-Wide Scale by integrating physico-chemical conditions with wetland biota. [19] developed an integrated wetland assessment tool for condition and health (WATCH) by incorporating horizontal position, vertical position, biology, hydrology, soil condition, and water chemistry.

Some countries, like Australia, created a system of environmental indicators for wetlands that took into account hydrology, riparian vegetation, salinity, and nutrient concentrations in surface and groundwater [20]. In Europe, the "Water Framework Directive of the European Union" proposed an ecological indicator system based on biological elements like flora, benthic invertebrates, and fishes, and hydromorphological and physical–chemical elements that support the biological elements [13]. The International Joint Commission (IJC) of Canada and the United States suggested 16 indicators composed of 41 measures (from physical, chemical, and biological indicator groups) to be the best indicators for assessing progress for Great Lakes protection and restoration efforts [21].

In the case of Ethiopia, monitoring of freshwater ecosystem health has so far focused on macroinvertebrates and, to a lesser extent, on diatoms [22]. In Ethiopia, the use of macroinvertebrates has increased in recent times [22–26]. Biomonitoring of freshwater ecosystem health using diatoms and plants as indicator species in East Africa, including Ethiopia, are very limited. [27,28] used diatoms to monitor the ecosystem health of rivers in Ethiopia, while [29] developed a multimetric plant-based index of biotic integrity for assessing the ecological state of forested, urban, and agricultural natural wetlands in Jimma Highlands, Ethiopia. However, an integrated wetland condition index is not yet developed for monitoring freshwater ecosystem health in Ethiopia.

Consequently, this research aimed to develop an integrated wetland condition index (IWCI) for lacustrine fringe wetlands (LFWs) in Lake Tana, Northwest Ethiopia based on characteristics and components that define wetlands, literature reviews, and the data collection and analysis of this research project. Wetlands surrounding Lake Tana have been severely impacted by recession agriculture, water hyacinth infestation, and short- and long-term water level fluctuations (WLFs), as well as a heavy sediment load. These lacustrine wetland stressors or drivers, individually and in combination, affect the overall health of Lake Tana lacustrine wetlands. Therefore, these impacts on lacustrine wetlands in Lake Tana need to be monitored and prioritized for future wetland restoration and conservation using holistic and integrated wetland conditions. The need for the integration of hydrology, wetland water chemistry, wetland sediment characteristics, and biota components for lacustrine wetland condition assessment is now especially urgent to assess the progress for Lake Tana protection and restoration efforts in general, and maintaining lacustrine wetlands for fish breeding ground and sediment control in particular. Our research helps to design rehabilitation plans, monitor and assess any future improvements in the conservation and restoration conditions of Lake Tana's lacustrine wetlands, and advise decision-makers about the priorities for lacustrine wetland conservation and restoration. The results of this study will also guide the establishment of lacustrine fringe wetlands monitoring sites and determine the types of parameters that shall be monitored.

## Materials and methods

### Lacustrine wetland site selection and sampling site selection

Six LFWs of Lake Tana were selected (Fig 1).The selection of sites as reference and impacted may be based on a prior knowledge of human disturbance impact pressures (e.g., hydrological modification, habitat alteration, buffer landscape and watershed disturbances, and invasive species) [24,30] or may involve a post classification based on human disturbance score of [31].This study used the former approach.

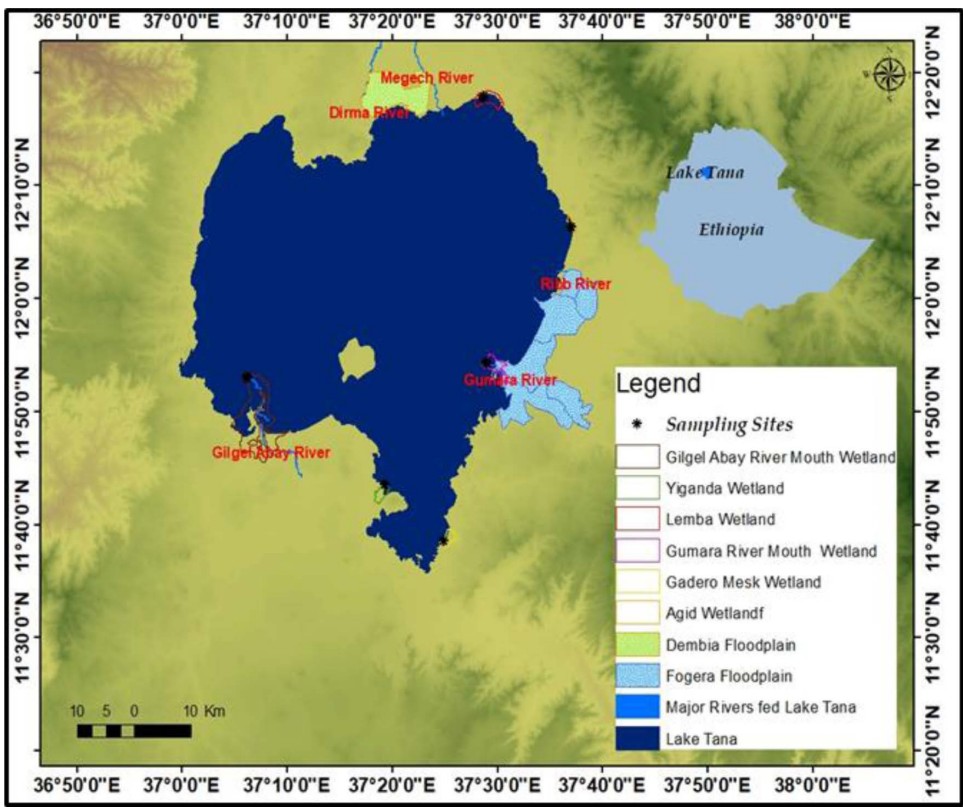

**Fig 1. Location map of six selected lacustrine fringe wetlands in Lake Tana.** *The study area of six selected LFWs in Lake Tana and three sampling sites in each selected wetlands using Ethiopian shape file, which includes Lake Tana, Major Rivers fed Lake Tana, Dembia Floodplain, and Fogera Floodplain.*

A.  *Less impacted wetlands:* [30] defined a reference site as a condition that is representative of a group of 'least impaired' sites that are characterized by selected physical, chemical, and biological characteristics. Given that natural reference wetlands were too difficult to find due to anthropogenic activity, Yiganda and Gadero wetlands were selected as reference wetlands (less impacted wetlands). Yiganda is less impacted than Gadero because Gadero is partly impacted by recessionary agriculture and the encroachment of plantations of eucalyptus trees. These lacustrine wetlands were categorized as least impacted (LI) provided they possessed intact riparian corridors, diverse ecosystems with submergent, floating, and emergent plant zones, and were visually appealing. There are no infestations of water hyacinth in these wetlands.

B.  *Highly impacted wetlands* These wetlands are severely impacted by recessionary agriculture, water hyacinth infestations, and short- and long-term water level fluctuations (WLFs). Both at Agid and Lemba wetlands, ecological restoration practices such as water hyacinth removal, dumping, prescribed burns, and other management factors useful for promoting native species biodiversity are being implemented.

C.  *River mouth-influenced wetlands:* Gumara River Mouth and Gilgel Abay River Mouth were chosen as wetlands impacted by river mouths. Because, besides lake water, these wetlands are hydrologically connected with major rivers that feed Lake Tana, namely: Gilgel Abay and Gumara. Rivers contribute a significant quantity of sediment to RI wetlands, which causes the Gumara and Gilgel Abay wetlands to create deltas. These river mouth wetlands have extremely shallow waters during the dry season as a result of being impacted by recessionary agriculture. Gumara

River Mouth wetland, unlike the Gilgel Abay River Mouth, it is invaded by water hyacinth. The predominant plant in the Gilgel Abay River wetland is papyrus (Cyperus papyrus).

## Methods of data collection

**Sampling protocols and procedures.** In this study, field samplings were carried out in the following order: (a) wetland water chemistry in-situ measurements; (b) wetland water sampling for laboratory nutrient analyses; (c) plankton sampling; (d) macrophyte sampling and (e) wetland sediment sampling [32]. A multiscale approach (between seasons, between wetlands, and among Ecophases within a given wetland) was applied. This study investigated the spatial patterns of macrophytes and plankton across six lacustrine wetlands in Lake Tana during two distinct hydrological phases: the pre-rainy season (drawdown phase or LWL) in May 2021 and the post-rainy season (post-flooding phase or PFP) in October 2021. Sampling for lacustrine wetland biota and environmental variables were carried out using the Protocol of [33] suggested four Ecophases. In this study, sampling of both environmental and lacustrine wetland biota data collection was conducted using hydrological gradients at each of three Ecophases: 1) Hydrophase (2-1m above bottom); 2) Littoral Ecophase (0.5–1 m above bottom); 3) Limosal Ecophase (from 0.1 m above to 0.5 m below soil surface). Hydrophase is nearby to limnetic zone; Littoral Ecophase middle position; and Limosal Ecophase as the edge of the lake or typically shoreline.

## Environmental data collection

**Water physicochemical variables.** Water chemistry field measurements and water samples were collected prior to any other sampling at each Ecophase. Before collecting water samples for laboratory chemistry analyses, the measurements of pH value, water temperature, electrical conductivity, TDS, and turbidity were made at in-situ physical measurements using an YSI Pro-30 Conductivity/TDS/Salinity/Temp Meter. Water samples were collected approximately 25 cm below the water's surface, directly above macrophyte beds at each site, concurrently with in-situ measurements.

For total nitrogen (TN) and total phosphorus (TP), 50 ml water samples were collected from each plot close to the water surface. Nutrients availability in the water column such as total phosphorus, and total nitrogen determined in the laboratory. Total phosphorus (TP) concentrations determined using PhosVer®3 (Hach Company, Loveland, CO, USA) based on the acid-per-sulfate digestion method in the range of 0.06–3.50 $PO_4^{3-}$ mg P L$-1$. Total nitrogen (TN) concentration also determined using the acid-per sulfate digestion method in the low range of 0.5–25.0 mg N L$-1$).The absorption was then measured using Hach's DR.2008 and DR.3900 spectrophotometers at a wavelength of 410 nm and 890 nm for TN and TP, respectively. Besides, to multiple hydrology regimes and sediment characteristics, wetland water chemistry included as confounding environmental factors to investigate both the effects of hydrology and sediment characteristics on wetland biota.

**Sediment characteristics.** Sediment samples were collected from the same locations as plant and water samples by grabbing a sediment sample at a depth of 60 cm. Pushing a 2-cm-diameter rod with a length of 2 meters labeled in cm and m into the substrate up to the end of the sediment determines the depth of the sediment. Placing the finger at this point of the substrate/water interface allows the rod to be withdrawn and labeled depths read.

Each sample was then placed in clean plastic zip-lock bags, carefully labeled, and transported for laboratory analysis in iceboxes. It is then oven-dried until constant weight at 60°C, crushed, and homogenized. These dried samples were reserved for organic matter (OM), sediment available for phosphorous analysis, and sediment pH. The OM of sediment samples was determined by drying them in a 105°C oven for two hours, weighing them, and then ashing them in a 360°C furnace for two hours. After ashing, the remaining sample material was cooled, weighed, and then passed through a 63-m sieve under running water. The Walkley and Black rapid titration method [34] was used to determine sediment OM, while the Olsen and Sommers method was used to determine total phosphorus [35].

**Hydrological regime measurement.** Two seasons that defined the wetland hydrology phases, such as the pre-rainy (drawdown phase) of May 2021 and the post-rainy (flooding phase) of October 2021, were studied. Four hydrological regimes, such as: (a) water depths in meters; (b) extent of water cover in hectares (ha); (c) durations of inundation in days; and (d) rates of recession or refill in m/year were included in this study. Lacustrine wetland water depth was measured at the center point of each quadrat with the aid of a 3-meter-long wooden pole. The Lacustrine wetland areas covered with water in ha in each of the 2 phases were determined using GPS readings from the end of the water levels recorded during field data collection and Moderate Resolution Imaging Spectroradiometer (*MODIS*) satellite images in May 2021 and October 2021. The extent of water cover has been calculated in ArcGIS Desktop 10.7.1 using GPS readings and MODIS satellite images as input. Rates of recession were measured by taking GPS readings from the maximum water level, while rates of refill were measured by taking GPS readings from the minimum water level. Finally, length in meters per year was calculated in ArcGIS Desktop 10.7.1 using GPS reading as an input.

## Lacustrine wetland biota sampling

In this study, lacustrine biotic response variables such as macrophytes, phytoplankton, and zooplankton were sampled and identified in the field as well as in the laboratory.

**Macrophyte sampling and lifeform assignment.** The quadrats were placed at each of the three Ecophases for each selected lacustrine wetland site. According to the sample plan, this arrangement enables the quadrats to follow the water-level history of each elevation. Only the aboveground biomass was taken into consideration, while submerged plants were collected using rakes. There are six selected Lacustrine wetland sites, and the total sample sites for the whole study area were 18 (3*6). Macrophyte sampling was conducted in each of the two seasons that defined the wetland hydrology phases, such as the pre-rainy (drawdown phase) of May 2021 and the post-rainy (flooding phase) of October 2021.

First, in each quadrat, the macrophyte species present were identified. For macrophyte species that could not be identified in the field, samples were placed in plastic bags with lacustrine wetland water and transported in coolers on ice to the laboratory for later identification. Having finished macrophyte specifies present identification, percent cover estimations were made using Braun-Blanquet scaling with corresponding degree of coverage and transformed metric scaling [36]: r= (1–2 individuals or very few individuals and 0.02), +=(<1% or few individuals and 0.1), 1=(1–5% or very rare and 2.5), 2m= (Many, but<5% and 5), 2a=(5–12.5% and 8.75), 2b= (12.5–25% and 18.75), 3=(25–50% or common and 37.5), 4=(50–75% or frequent and 62.5) &5=(75–100% or predominantly abundant and 87.5) and transformed metric scaling.

To describe different forms of exploiting resources in relation to the hydrological conditions, which might indicate different functions in an aquatic ecosystem, according to [37] in this study macrophytes categorized into six different life forms as proxies of functional groups: (i) emergent (Em) species; (ii) rooted floating-stemmed (RFS) species; (iii) rooted floating-leaved (RFL) species; (iv) free floating (FF) species; (v) free submerged (FS) species; and (vi) rooted submerged (RS) species.

**Plankton sampling and laboratory analysis.** In the case of phytoplankton sampling, from sites of "*Hydrophase and Littoral Ecophases*", samples were collected using a mesh size of 25µm phytoplankton net. From sites of "*Limosal Ecophases*," samples were collected with a bucket by filtering 20L of water using a mesh size of 25µm phytoplankton net. Finally, Samples were labelled and preserved using Lugol's Iodine Solution. Of this concentrated sample, 1 ml sub-sample was transferred into the Sedgwick-Rafter cell, which had 1,000 grids, after being fully homogenized by gentle inversion and agitation. Similarly for zooplankton sampling, from sites of "*Hydrophase and Littoral Ecophases*", samples collected with a Juday net with a mesh size of 64 µm from the bottom to the surface. From sites of "*Limosal Ecophases*," samples collected with a bucket by filtering 50L of water through a 64-µm mesh. Samples were labelled and preserved using Alcohol 85%.

Phytoplankton was identified to the species level using taxonomic [38].Using the necessary taxonomic literature [39,40], zooplankton was identified to the species level under a stereoscopic microscope at ×400 magnification. Rotifers

and Cladocerans were classified as a single age class, while calanoids and cyclopoids were counted according to their developmental stages (adults, copepodites, and nauplii).

## Data analysis

**Wetland hydrology, water, and sediment parameter analysis.** Hydrology, water, and soil parameters within three wetland impact level categories were compared using Friedman post-hoc, Fisher's LSD pair wise comparison in R version 4.1.3(2022-03-10). Fisher's LSD pairwise comparison method was chosen in this study for the following reasons: (a) Fisher's LSD is essentially a series of individual t tests, in contrast to the Bonferroni, Tukey, Dennett, and Holm methods; instead of computing the pooled SD from just the two groups under comparison, this makes it more difficult to achieve significance; instead, it computes the pooled SD from all the groups, which increases power; (b) Fisher's LSD method for multiple comparisons is used in ANOVA to create confidence intervals for all pairwise differences between factor level means, whereas Dunn's test is the appropriate nonparametric pairwise multiple-comparison procedure when a Kruskal–Wallis test is rejected; and (c) Fisher's LSD test implemented in the R software using the "agricolae" package, whereas Dunn's test is implemented for Stata in the dunntest command.

The Friedman test was performed using the R package "agricolae," and when significant, we used Fisher's least significant difference (LSD) test for pairwise post hoc comparisons implemented in the same package. For complete functionality of agricolae package, other packages such as MASS, nlme, klaR, Cluster, and algDesign were installed and loaded. Our data is organized in a two-way table with n rows and K columns (impact level categories represent various conditions/treatments and values of each hydrology, water, and sediment parameter).

**Indicator species selection and analysis.** Macrophyte, and plankton indicator species tests (IndVal% and p (raw)) were calculated using the macrophyte, and plankton community dataset in PAST 4.14 (PAleontological STatistics) [41]. Indicator Species Analysis (ISA) requires a priori groups and data on the abundance or presence of individuals in each group. Thus, wetlands were grouped into less impacted wetlands, river mouth-influenced wetlands, and highly impacted wetlands. This ensured that the pool of least impacted wetlands were used to define sensitive species, whereas the pool of more impacted wetlands, were used to define tolerant indicator species. The 'Analysis of Similarities' (ANOSIM) procedure in PAST4.14 was then carried out to test for differences between macrophyte, phytoplankton, and zooplankton assemblages among three impact categories of wetlands (less impacted wetlands, river mouth-influenced wetlands, and highly impacted wetlands), analyzing the pre-rainy and post-rainy datasets separately. The 'Similarity Percentages' (SIMPER) procedure is performed in PAST 4.14 to quantify taxa contributing to the average Bray-Curtis dissimilarity among the three impact categories of wetlands.

**Description and calculation of wetland condition index.** The index of lacustrine wetland condition was developed based on four wetland characteristics that define wetlands: hydrology, wetland water physiochemical environment, wetland soil environment, and wetland biota.

**Scoring of wetland hydrology, water chemistry, and sediment characteristics.** The value obtained from field and laboratory measurements of each wetland hydrology, water chemistry, and sediment characteristics indicator was normalized by allocating scores based on (a) literature information and (b) the author's field on relatively undisturbed reference wetlands (S1 and S9 Tables).

(a) *Water regime index (WRI):* Hydrologic conditions are critical for maintaining the structure and functionality of a wetland. Wetland conditions could thus virtually certainly alter as a result of changes in hydrology [2]. A change in hydrology can result from a change in any of the hydrological components: the depth of inundation (m), the area of inundation (ha), the duration of inundation (days), and the rates of recession and refill in m/month. Persistent shifts in the timing of either the seasonal maximum or minimum may reflect shifts in the regional water budget and provide insight into potential impacts on aquatic plants and fish spawning habitats, as well as other sensitive aspects of the lacustrine ecosystem.

(b) *Water quality index (WQI):* Due to the connection between water properties and the ecological processes of a wetland, changes in a wetland's physico-chemical components are likely to result in changes in the wetland's state [2]. Many of the biotic elements of wetlands' processes (e.g., feeding, growth, and reproduction of fauna and growth of flora) will be impacted by changes to water characteristics. This suggested integrated index of lacustrine wetland condition takes into account the key water characteristics of the wetland, including electrical conductivity, total dissolved solids, pH, water temperature, total nitrogen, and total phosphorous.

(c) *Sediment quality index (SQI):* As soils play an important role in nutrient storage, wetland plant growth, seed storage, and habitat for benthic aquatic animals and soil microorganisms, changes to wetland soils are expected to affect the condition of wetlands [2].This suggested integrated index of lacustrine wetland condition takes sediment pH, organic matter, accessible phosphorous, substrate type, and substrate depth into account as the primary sediment parameters of the wetland.

**Scoring of wetland biota index (WBI) or indicator species.** The concept that cumulative effects of environmental changes are integrated over, or reflected by, the current status or trends (short- or long-term patterns of change) in the diversity, abundance, reproductive success, or growth rate of one or more species living in that environment is the foundation of using indicator species [42–44]. Macrophyte, phytoplankton (diatom), and zooplankton are wetland biotas included in this lacustrine wetland index development.

Macrophyte, diatom, zooplankton and macrophyte IWCI in Lake Tana were scored using the following steps (S10 Table): (a) Calculation values for the three metrics (percent tolerant indicator species, percent sensitive indicator species, and percent exotic species); (b) taking the natural log of metrics to improve distribution (ln (metric value + 10)). For wetland status, we used the straight metric values (without using the natural log); (c) we used the scoring equations to normalize scores between 0 and 10 using [45] equation, as follows:

Metrics that increase with increasing wetland impact level – tolerant, exotic metrics of macrophytes, pH and Class 3 of diatom phytoplankton:

$$= 10 - ((\text{metric-5th percentile}) * (10/ (\text{95th percentile-5th percentile}))) \tag{Eq. 1}$$

Metrics that decrease with increasing wetland impact level- sensitive metric:

$$= ((\text{metric-5th percentile}) * (10/ (\text{95th percentile-5th percentile}))) \tag{Eq. 2}$$

where values less than 0 are interpreted as 0, and values more than 10 as 10 (S11 Table).

**Overall wetland condition index calculation and validation**

The values obtained from field and laboratory measurements of each indicator were normalized by allocating scores based on a literature review of wetland condition indicators and the results of this research project from relatively undisturbed reference wetlands and different disturbance levels in shore area wetlands in Lake Tana. Scores were allocated on a scale from 1 to 10, in such a way that the highest scores reflected the best conditions and the lowest scores the most disturbed and unhealthy conditions. Wetland condition indicators in each of the four wetland components were determined based on the above-mentioned data analysis of Fisher's least significant difference (LSD), indicator species analysis, 'Analysis of Similarities' (ANOSIM) results, and 'Similarity Percentages' (SIMPER) analysis results.

Within four sub-indexes, there are more than one measure, with each measure given equal weighting, whereas the four-wetland components (hydrology, wetland water physiochemical environment, wetland soil physiochemical environment, and biota) are weighted as per their contribution to the overall wetland condition. Weights were assigned using the expected value or Ranking Method using the Decision Support System Software of DEFINITE [46].

The overall ILWCI is calculated by multiplying each sub-index score by its respective final weight and summing the scores. This is represented by the formula:

$$\mathbf{ILWCI}_{total} = \mathbf{WRI}_w * \mathbf{WRI}_{si} + \mathbf{WQI}_w * \mathbf{WQI}_{si} + \mathbf{SQI}_w * \mathbf{SQI}_{si} + \mathbf{WBI}_w * \mathbf{WBI}_{si}$$ (Eq. 3)

$$\mathbf{ILWCI\ total} = \sum_{si=0}^{4} (\mathbf{wsi})$$ (Eq. 4)

Where ILWCI total is the total IWC score,

*w* is the final weight of the corresponding sub-index, and

*si* is the sub-index score

Finally, the total IWCI was classified into five wetland condition categories, namely: very poor or very far from the reference (0–3), poor (3–5), moderate (5–7), good (7–9), and excellent wetland condition or reference condition (9–10). Correct classification percentage (CCP) [47] used to measure precision. A CCP higher than or equal to 70% is considered as reliable in ecological applications [48].

## Results

### Hydrology, water and soil parameters

Hydrological regimes of 5 parameters, water chemistry of 7 parameters, and 5 sediment parameters of 18 samples from six wetlands were analyzed for lacustrine wetlands in Lake Tana. Table 1–2 showed mean values, standard deviation, 95% confidence interval, group, and least significant difference for hydrological regimes, water, and sediment parameters for three categories (less impacted wetlands, river mouth influenced wetlands, and highly impacted and ecological restoration wetlands).

Fisher's LSD pairwise comparison showed no significant difference between means with similar letters. Of the 17 hydrological regimes, water and sediment parameters measured during low water levels, inundated, exposed, and rates of recession from hydrology parameters, water temperature, turbidity, and phosphorus from water parameters, available phosphorus, and substrate size from sediment parameters were not significantly different among the three impact categories (Table 1). During low water levels, three hydrological regimes, four water parameters, and two sediment parameters showed a significant difference ($p < 0.05$) among less impacted wetlands, river mouth-influenced wetlands, and highly impacted wetlands. Less-impacted wetlands had significantly different Inundated (x = 31.90 ± 4.95) and p = 1e-04***, Exposed (x = 68.11 ± 4.95) and p = 1e-04***, rates of recession (x = 688.50 ± 291.4) and p = 0.0000***, water pH (x = 8.12 ± 0.45) and p = 0.0002***, EC (x = 0.15 ± 0.01) and p = 0.0113*, TDS (x = 95.25 ± 4.43) and p = 0.0070**, sediment depth (x = 53.33 ± 50.91) and p = 0.0266*, SOM(x = 8.99 ± 5.31) and p = 0.0013** than river mouth-influenced wetlands (Table 1). Inundated (x = 62.03 ± 1.88) and p = 0e + 00***, Exposed (x = 37.98 ± 1.88) and p = 1e-04***, rates of recession (x = 611.50 ± 70.66) and p = 0.0000***, water pH (x = 7.82 ± 0.42) and p = 0.0023**, EC (x = 0.16 ± 0.05), and TDS (x = 107.97 ± 22.46) and p = 0.0007***, water TN (x = 2.36 ± 0.16), and p = 0.0000***, and sediment depth (x = 39.13 ± 26.34) and p = 0.0068*** were also significantly different in highly impacted wetlands than in river mouth-influenced wetlands. Only five parameters of inundated (p = 0e + 00***,) and exposed (p = 0e + 00***), water TN (p = 0.0000***), sediment pH (p = 0.0192*), and SOM (p = 0.0023**) were significantly different between less impacted and highly impacted wetlands (Table 1).

Of the five hydrological parameters measured during high water level, only exposed, and rates of refill were significantly different among the three impact categories. Of the seven water parameters measured during the post-flood period, except for water TP, the rest six parameters (water PH, water temperature, EC, TDS, turbidity, and water TN) were not

**Table 1. Hydrology, water and soil parameters among three impact categories of wetlands, including less impacted wetlands, river mouth influenced wetlands, and highly impacted lacustrine wetlands during low water level.**

| Parameter | Less Impacted (LI) | | | River Mouth Influence (RMI) | | | Highly Impacted (HI) | | |
|---|---|---|---|---|---|---|---|---|---|
| | X±σ | 95%CI | G | X±σ | 95%CI | G | X±σ | 95%CI | G |
| **Hydrology Parameter** | | | | | | | | | |
| Depth(m) | 0.72±0.44 | 0.27 to 1.16 | a | 0.59±0.62 | 0.27 to 1.16 | a | 0.43±0.46 | 0.88 | a |
| Inundated (%) | 31.90±4.95 | 28.76 to 35.03 | c | 43.49±3.31 | 40.36 to 46.62 | b | 62.03±1.88 | 58.89 to 65.16 | a |
| Exposed (%) | 68.11±4.95 | 64.96 to 71.25 | a | 56.46±3.36 | 59.61 to 59.61 | b | 37.98±1.88 | 34.83 to 41.12 | c |
| Duration(day) | 344.17±32.31 | 317.00 to 371.34 | a | 346.83±28.92 | 374.00 to319.66 | a | 344.17±32.31 | 317.00 to 371.34 | a |
| Rates of Rec.(m/yr) | 688.50±291.4 | 520.18 to856.82 | b | 2846.50±148.43 | 2678.18 to3014.82 | a | 611.50±70.66 | 443.18 to779.82 | b |
| **Water Parameter** | | | | | | | | | |
| pH | 8.12±0.45 | 7.76 to 8.47 | a | 6.95±0.36 | 6.59 to 7.31 | b | 7.82±0.42 | 7.47 to 8.18 | a |
| Temp(0c) | 26.85±2.69 | 23.76 to 29.94 | a | 24.88±3.35 | 21.80 to 27.97 | a | 25.70±4.39 | 22.61 to 28.79 | a |
| EC(mS/cm) | 0.15±0.01 | 0.12 to 0.19 | a | 0.08±0.05 | 0.05 to 0.05 | b | 0.16±0.05 | 0.13 to 0.20 | a |
| TDS(mg/L) | 95.25±4.43 | 78.57 to111.93 | a | 60.72±24.06 | 24.06 to 44.03 | b | 107.97±22.46 | 91.28 to 124.65 | a |
| Turbidity | 10.25±6.42 | −2.26 to 22.76 | a | 32.28±11.36 | 11.36 to 19.76 | a | 23.25±21.22 | 10.74 to 35.76 | a |
| TN(mg/L) | 4.47±0.52 | 4.07 to 4.86 | a | 4.49±0.57 | 4.09 to 4.89 | a | 2.36±0.16 | 1.96 to 2.76 | b |
| TP(mg/L) | 1.26±0.35 | 0.75 to 1.78 | a | 1.46±0.75 | 0.94 to 1.97 | a | 1.06±0.59 | 0.55 to 0.55 | a |
| **Sediment Parameter** | | | | | | | | | |
| pH | 6.42±0.41 | 6.75 to 7.97 | b | 7.36±1.05 | 5.81 to 7.02 | ab | 7.47±0.45 | 6.86 to 8.08 | a |
| Depth(cm) | 53.33±50.91 | 21.76 to 84.91 | ab | 104.83±25.79 | 25.79 to 73.26 | a | 39.13±26.34 | 7.56 to 70.71 | b |
| SOM (%) | 8.99±5.31 | 6.12 to 11.86 | a | 1.50±0.87 | −1.37 to 4.37 | b | 2.02±1.95 | −0.85 to 4.89 | b |
| SAP(ppm) | 7.41±4.74 | 1.76 to13.06 | a | 9.98±6.87 | 4.33 to 15.63 | a | 9.13±7.55 | 3.47 to 14.78 | a |
| SS(mm) | 2.33±1.97 | 1.08 to 3.59 | a | 1.01±1.54 | −0.25 to 2.26 | ab | 0.07±0.10 | −1.18 to 1.33 | b |

X±σ values represent the mean±standard deviation; 95%CI = 95% confidence interval; G = group; and LSD = Least significant difference. Categories with dissimilar letters were significantly different (Fisher's LSD pair wise comparison, α = 0.05)

significantly different among the three impact categories. From the four sediment parameters, sediment pH and sediment available phosphorus were not significantly different among the three impact categories (Table 2). Only two hydrological, one water and two sediment parameters showed a significant difference (p < 0.05) among less impacted wetlands, river mouth influenced wetlands, and highly impacted wetlands. Highly impacted wetlands had significantly different exposed area (x = 10.92±0.42) and p = 0.0000*** , water TP (x = 2.14±0.07) and p = 0.0069** than less impacted wetlands of Exposed(x = 18.72±2.98), rates of refill(x = 463.00±249.76), water TP (X = 2.95±0.58). Less impacted wetlands had significantly different exposed (x = 18.72±2.98) and p = 0.0000*** , SOM (x = 4.77±0.72) and p = 0.0100** than river mouth-influenced wetlands less wetlands of SOM (X = 2.81±1.34) (Table 2).

### Indicator species analysis

**Macrophyte indicator species.** During the low water level period, using indicator analysis, *Ceratophyllum demersum* (IndVal = 50% and p = 0.0233) and *Persicaria senegalensis* (IndVal = 50% and p = 0.0226) were indicators of the less impacted wetland sites, while *Cyperus rotundus* was significant (IndVal = 50% and p = 0.0246) in highly impacted wetlands. No indicator species were found for river mouth-influenced wetland sites. During the high water level period, only *Eichhornia crassipes* was a significant (IndVal = 67.83% and p = 0.0008) indicator of the highly impacted wetland sites, whereas in both river mouth-influenced and less impacted wetland sites, no indicator species were found (S2 Table and S1 Fig.).Tolerant indicator and invasive exotic species increased with increasing wetland anthropogenic impact level,

**Table 2. Hydrology, water and soil parameters among three impact categories of wetlands, including less impacted wetlands, river mouth influenced wetlands, and highly impacted lacustrine wetlands during high water level.**

| Parameter | Less Impacted (LI) | | | River Mouth Influence (RMI) | | | Highly Impacted (HI) | | |
|---|---|---|---|---|---|---|---|---|---|
| | X ± σ | 95%CI | G | X ± σ | 95%CI | G | X ± σ | 95%CI | G |
| **Hydrology Parameter** | | | | | | | | | |
| Depth(m) | 0.75±0.61 | 0.15 to 1.34 | a | 0.86±0.84 | 0.27 to 1.46 | a | 0.66±0.56 | 0.06 to1.25 | a |
| Inundated (%) | 81.28 ±2.98 | 79.37±83.19 | a | 10.46±2.33 | 8.55 to 12.37 | a | 89.08±0.42 | 87.63 to 91.45 | a |
| Exposed (%) | 18.72±2.98 | 16.81±20.63 | a | 10.46±2.33 | 8.55 to 12.37 | b | 10.92±0.42 | 9.01 to 12.83 | b |
| Duration(day) | 235.67±142.05 | 112.54±358.80 | a | 230.17±147.80 | 107.04 to353.30 | a | 192.50±134.34 | 69.37 to 315.63 | a |
| Rates of Ref.(m/yr) | 463.00±249.76 | 153.67±772.33 | b | 1156.50±560.32 | 847.17 to 1465.83 | a | 285.00±52.58 | −24.33 to 594.33 | b |
| **Water Parameter** | | | | | | | | | |
| pH | 8.04±0.38 | 7.48 to8.60 | a | 8.76±0.79 | 8.20 to 9.32 | a | 8.06±0.68 | 7.50 to8.62 | a |
| Temp(0c) | 22.22±2.25 | 19.56 to24.87 | a | 24.30±0.88 | 21.64 to26.96 | a | 22.15±4.70 | 19.49 to24.81 | a |
| EC (mS/cm) | 0.14±0.02 | 0.12 to 0.17 | a | 0.12±0.02 | 0.10 to 0.15 | a | 0.14±0.03 | 0.12 to0.16 | a |
| TDS(mg/L) | 99.80±17.18 | 65.90±94.34 | a | 80.12±17.29 | 85.58 to114.02 | a | 95.50±14.39 | 81.28 to109.72 | a |
| Turbidity (NTU) | 86.53±179.29 | −30.78 to 203.84 | a | 53.27±179.29 | −64.04 to 170.58 | a | 155.92±123.37 | 38.61 to 273.23 | a |
| TN (mg/L) | 3.76±0.62 | 2.93 to4.59 | a | 4.11±1.52 | 3.28 to 4.93 | a | 3.61±0.14 | 2.78 to 4.44 | a |
| TP(mg/L) | 2.95±0.58 | 2.56 to 2.56 | a | 2.61±0.50 | 2.22 to 3.00 | ab | 2.14±0.07 | 0.50 to2.53 | b |
| **Sediment Parameter** | | | | | | | | | |
| pH | 5.74±0.30 | 6.06 to 6.04 | a | 5.63±0.27 | 5.33 to5.93 | a | 5.76±0.44 | 0.27 to 6.06 | a |
| Depth(cm) | 23.67±13.65 | 12.85 to 34.48 | b | 41.67±15.06 | 30.85 to 52.48 | a | 22.67±7.12 | 11.85 to 33.48 | b |
| SOM (%) | 4.77±0.72 | 3.77to5.77 | a | 2.81±1.34 | 1.80 to 3.81 | b | 4.27±1.29 | 1.29 to3.27 | ab |
| SAP(ppm) | 18.60±3.83 | 13.57 to 23.63 | a | 14.10±4.49 | 9.07to 19.13 | a | 20.97±3.83 | 15.93 to26.00 | a |
| SS(mm) | 0.87±2.04 | −0.69 to 2.36 | a | 1.00±1.55 | −0.52 to 2.53 | a | 1.50±1.64 | −0.02 to 3.03 | a |

whereas sensitive and wetland status indicator species displayed the opposite trend. *Cyperus rotundus* was the only tolerant indicator species of highly impacted wetlands, whereas *Ceratophyllum demersum* and *Persicaria senegalensis* were identified as sensitive indicator species of the less impacted wetland sites. Only two invasive exotic species were found in sampling sites of LFWs in Lake Tana, all including *Azolla fern* and *Eichhornia crassipes. Cyperus digitatus, Cyperus papyrus, Echinochloa species, Nymphaea sp., Potomagton pectinatus,* and *Typha latifolia* were considered as wetland status species (including both obligate and facultative wetland species). Both *Azolla fern* and *Eichhornia crassipes* were found at (Gumara River Mouth, Gadero and Lemba).

The ANOSIM tests showed significant differences in macrophyte assemblages among three impact categories of wetlands (less impacted wetlands, river mouth-influenced wetlands, and highly impacted wetlands) during both the pre-rainy (R=0.1971, p=0.0073) and post-rainy seasons (R=0.393, p=0.0005). Results from SIMPER analyses indicated that the overall average Bray-Curtis dissimilarities between all pairs of sites in LI and HI wetlands were 100% in both pre-rainy season and post-rainy season and were made up mainly of contributions from *Potomagton pectinatus (*19.49%) and *Eichhornia crassipes (*45.13%), respectively. The overall average Bray-Curtis dissimilarity between LI and RMI was 86.71% in the pre-rainy season and 80.01% in the post-rainy season and was made up mainly of contributions from *Potomagton pectinatus (*21.48%) and *Echinochloa species (*26.6%), respectively. The overall average Bray-Curtis dissimilarity between HI and RMI was 90.08% in the pre-rainy season and 83.99% in the post-rainy season and were made up mainly of contributions from *Eichhornia crassipes (*22.59%*)* and *Eichhornia crassipes (*37.93%), respectively (S3Table).

**Diatom indicator species.** Diatoms have become the most commonly used indicator group because they are taxonomically different, abundant in almost all aquatic environments, and respond quickly to changing conditions. Thus, from four taxonomic groups of phytoplankton identified in this study, diatom species were included for the development of IWCI for LFWs in Lake Tana.

During the pre-rainy season, twenty-five diatom species were identified, while during the post-rainy season, ten diatom species were identified (S4–S5 Tables and S2 Fig).Using species-level abundance data for the diatom assemblage, indicator species analysis (ISA) was used to establish the lists of tolerant and sensitive indicator species. In addition to indicator species calculation using PAST software, autecological metrics based on available research that correlated individual diatoms with morphology, behavior, and the physical and chemical water environment were also applied in this study. From different autecological metrics, pH Class 3 (circumneutral, mainly occurring at pH values around 7) [49] was used in diatom-based metrics calculation. Tolerant Diatom Indicator Species and pH Class 3 (circumneutral, mainly occurring at pH values around 7) Diatom Indicator Species increased with increasing wetland anthropogenic impact level, whereas sensitive diatom indicator species displayed the opposite trend.

Using indicator analysis, during the low water level period, *Aulacoseira granulata* (IndVal=97.58% and p=0.0006), *Cymbella sp.* (IndVal=66.67% and p=0.0049), *Diploneis parma* (IndVal=33.33% and p=0.0975), *Epithemia argus* (IndVal=50%% and p=0.0233), *Navicula placenta* (IndVal=33.33% and p=0.0975), *Navicula pseudotuscula* (IndVal=33.33% and p=0.0975), *Nitzschia dissipata* (IndVal=32.36% and p=0.0928), *Pinnularia sp.* (IndVal=33.33% and p=0.0975), *Surirella linearis* (IndVal=33.33% and p=0.0975), *and Synedra sp. (IndVal=99.72% and p=0.0003)* were identified as sensitive diatom indicator species of the less impacted wetland sites. The two Tolerant diatom indicator species in highly impacted wetlands with the highest indicator values included *Navicula sp.* (IndVal=78% and p=0.0478) and *Pleurosigma angulatum* (IndVal=33.33% and p=0.0905). *Cyclotella sp.* (IndVal=33.33% and p=0.0943) was the only diatom indicator species in river mouth-influenced wetland sites (S4 Table). During the high water level period, in highly impacted wetland sites, no indicator species were found. *Cymbella sp.* (IndVal=35.71% and p=0.0861) was sensitive diatom indicator species of the less impacted wetland sites both during low water level and high water level periods. *Gomphonema sp.* (IndVal=25.42% and p=0.0975) was a sensitive diatom indicator species of the less impacted wetland sites during the high water level period (S5 Table).

Under pre-rainy season, the ANOSIM tests revealed significant differences in diatom assemblages among the three impact categories of wetlands (river mouth-influenced wetlands, highly impacted wetlands, and less impacted wetlands) (R=0.156, p=0.0439). According to the results of SIMPER analysis, the overall average Bray-Curtis dissimilarities between all pairs of sites in LI and HI wetlands was 89.46% in pre-rainy season and 85.53% in post-rainy season, and was made up mainly of contributions from *Aulacoseira granulata* (65.57%) and *Aulacoseira granulata (*81.92%) respectively. The overall average Bray-Curtis dissimilarity between LI and RMI was 89.38% in pre-rainy season and 79.44% in post-rainy season and was made up mainly of contributions from *Aulacoseira granulata* (71.97%) and *Aulacoseira granulata (*60.29%) respectively. The overall average Bray-Curtis dissimilarity between HI and RMI was 89.76% in pre-rainy season and 85.07% in post-rainy season and was made up mainly of contributions from *Aulacoseira granulata* (30.77%) and *Aulacoseira granulata (*81.42%) respectively (S6 Table*)*.

**Zooplankton indicator species.** Using indicator analysis, during the high water level period, *Ceriodaphnia sp.* (IndVal=55.34% and p=0.0176), and *Thermocyclops* (IndVal=60.63% and p=0.0377), were identified as significant sensitive zooplankton indicator species of the less impacted wetland sites. The five tolerant zooplankton indicator species in highly impacted wetlands with the highest indicator values included *Bosmina longirostris* (IndVal=78% and p (raw) = 0.0226), *Brachionus patulus* (IndVal=64.09% and p=0.0021), *Brachionus sp.* (IndVal=41.03% and p=0.0484), *Keratella tropica* (IndVal=58% and p=0.0284), and *Mesocyclops* (IndVal=43.94% and p=0.082). No zooplankton indicator species were found for river mouth-influenced wetland sites (S7 Table and S3 Fig.*)*.

The ANOSIM tests showed there is no significant differences in zooplankton assemblages among three impact categories of wetlands (less impacted wetlands, river mouth-influenced wetlands, and highly impacted wetlands), during

post-rainy season (R = 0.09835, p = 0.1058). Results from SIMPER analyses indicated that the overall average Bray-Curtis dissimilarities between all pairs of sites in LI and HI wetlands was 82.54% in pre-rainy season, and was made up mainly of contributions from *Bosmina longirostris* (31.53%) and *Chydorus sp.* (17.73%). The overall average Bray-Curtis dissimilarity between LI and RMI was 85.03% in pre-rainy season and was made up mainly of contributions from *Chydorus sp.* (28.6%) and *Thermocyclops* (22.15%). The overall average Bray-Curtis dissimilarity between HI and RMI was 88.24% in pre-rainy season and was made up mainly of contributions from *Bosmina longirostris* (28.37%) and *Thermocyclops* (17.72%) (S8 Table).

**Overall wetland condition index.** The final IWCI of the LFWs resources in Lake Tana was computed by aggregating the four sub-indices into a single value. During low water level period, the IWCI varied from 5.50 to 9.22. The lowest threshold (5.50) represents moderate condition whereas the upper threshold (9.22) represents the most reference condition. During high water level period, the IWCI varied from 2.00 to 7.89. The lowest threshold (2.00) represents very poor or the most impaired condition whereas the upper threshold (7.89) represents good or mildly impacted condition (Table 3).

According to IWCI of low water level period, eight wetland sampling sites were found within moderately impacted condition (44.44%), similarly eight wetland sampling sites were found within good or mildly impacted condition, and only two wetland sampling sites(11.11%) were found with in excellent or reference condition. No wetland sampling sites were found within both very poor and poor condition. Among the 18 samples only 6 were correctly classified by the integrated wetland index (CCP = 33.33%). The CCP value of this study is far below 70%. This indicates that IWCI of low water level period is not considered as reliable in ecological applications [48].

IWCI of high water level period indicated that only 1 wetland sampling site (5.55%), 8 wetland sampling sites (44.44%), 4 wetland sampling sites (22.22%), and 5 wetland sampling sites (27.77%) were within very poor, poor, moderate and good wetland condition categories respectively. No wetland sampling sites were found within excellent or reference

**Table 3. Classification of sites into different levels of wetland condition based on Overall Wetland Condition Index scores in the study wetlands.**

| Sampling Site | Impact Categories | Low Water Level Period | | High Water Level Period | |
|---|---|---|---|---|---|
| | | IWCI | Categories | IWCI | Categories |
| 1GiA-Hyd | RMI | 5.58 | Moderate | 7.28 | Good |
| 2GiA-Lit | RMI | 5.80 | Moderate | 5.71 | Moderate |
| 3GiA-Lim | RMI | 8.99 | Good | 7.59 | Good |
| 4Yig-Hyd | LI | 8.48 | Good | 5.42 | Moderate |
| 5Yig-Lit | LI | 9.22 | Excellent | 7.30 | Good |
| 6Yig-Lim | LI | 8.62 | Good | 5.34 | Moderate |
| 7Gad-Hyd | LI | 8.33 | Good | 7.89 | Good |
| 8Gad-Lit | LI | 8.36 | Good | 7.89 | Good |
| 9Gad-Lim | LI | 8.41 | Good | 5.79 | Moderate |
| 10Gum-Hyd | RMI | 6.39 | Moderate | 4.88 | Poor |
| 11Gum-Lit | RMI | 8.58 | Good | 2.00 | Very poor |
| 12Gum-Lim | RMI | 9.04 | Excellent | 3.59 | Poor |
| 13Agi-Hyyd | HI | 6.20 | Moderate | 4.64 | Poor |
| 14Agi-Lit | HI | 5.87 | Moderate | 4.06 | Poor |
| 15Agi-Lim | HI | 5.94 | Moderate | 4.27 | Poor |
| 16Lem-Hyd | HI | 5.80 | Moderate | 4.21 | Poor |
| 17Lem-Lit | HI | 8.41 | Good | 4.36 | Poor |
| 18Lem-Lim | HI | 5.50 | Moderate | 4.43 | Poor |

condition (Table 3 and 4). Among the 18 samples only 10 were correctly classified by the integrated wetland index (CCP = 55.55%). The CCP value of this study is below 70%. This indicates that IWCI of high water level period is not considered as reliable in ecological applications [48]. IWCI does not discriminate reference from impaired sites. However as compare to IWCI of low water level period, IWCI of high water level period is better to discriminated the impacted wetland categories from the reference wetland categories. The selection of sites, as reference based on a prior knowledge of human disturbance impact pressures could not identify the exact reference sites. Thus, a post classification based on human disturbance score of [31] might could select the exact reference sites. Therefore, for future research works on the development of wetland condition index, reference and impacted sites needs to be selected based on a post classification approach than a prior knowledge of human disturbance impact pressures.

In both low water level and high water level periods, IWCI of LFWs in Lake Tana scores decreased with increasing anthropogenic wetland impact levels. In the beginning of this study, Agid and Lemba were selected as highly impacted wetlands because these wetlands are severely impacted by recessionary agriculture, water hyacinth infestations, and short- and long-term water level fluctuations (WLFs).Particularly during high water level period, all sampling sites (13Agi-Hyd, 14Agi-Lit, 15Agi-Lim, 16Lem-Hyd, 17Lem-Lit, and 18Lem-Lim) received scores of poor wetland condition as expected. Yiganda and Gadero wetlands were selected as reference wetlands (less impacted wetlands).Yiganda is less impacted than Gadero because Gadero is partly impacted by recessionary agriculture and the encroachment of plantations of eucalyptus trees. There are no infestations of water hyacinth in these wetlands. As oppose to the expectation, during high water level period from these less impacted wetland groups, no wetland sampling sites were found within excellent or reference condition while during low water level period only one sampling site (5Yig-Lit) of Yiganda wetland was found within excellent or reference condition (Table 3). Gumara River Mouth wetland, unlike the Gilgel Abay River Mouth, it is invaded by water hyacinth. During high water level period, all three wetland-sampling sites of Gumara River Mouth wetland coded as: 10Gum-Hyd, 11Gum-Lit, and 12Gum-Lim received scores of poor, very poor and poor wetland condition respectively as expected. Similarly, all three wetland-sampling sites of Gilgel Abay River Mouth wetland coded as: 1GiA-Hyd, 2GiA-Lit, and 3GiA-Lim got scores of good, moderate, and good wetland condition respectively as expected. During low water level period, all three wetland-sampling sites of Gumara River Mouth wetland coded as: 10Gum-Hyd, 11Gum-Lit, and 12Gum-Lim received scores of moderate, good, and excellent wetland condition respectively as oppose to our expectation.

Ranking of alternative IWCI of Low water level and high water level periods using multicriteria analysis of DEFINITE Software entailed the evaluation of the overall scores of alternative IWCI of Low water level and high water level periods in terms of Water Regime Index (WRI), Water Quality Index (WQI), Sediment Quality Index (SQI), and Wetland Biota Index. Weight Values for each sub-indices using Ranking Method and the contributions of each sub-indices in each alternative IWCI of Low water level and high water level periods (Fig 2b) showed that 52.10% of the weight was assigned to Wetland Biota Index followed by 27.10% and 14.60% to Water Quality Index (WQI), and Sediment Quality Index (SQI) respectively whereas, Water Regime Index (WRI) were assigned the least weights of 6.30%. Using these weights, during Low water

Table 4. Number of wetland sampling sites and percentage share of IWCI in each wetland condition categories.

| Range of IWCI | Wetland condition categories | Low Water Level Period | | High Water Level Period | |
|---|---|---|---|---|---|
| | | Number of Samples | % Share | Number of Samples | % Share |
| 0–3 | Very poor or Degraded | 0 | 0% | 1 | 5.55% |
| 3–5 | Poor or Moderately Degraded | 0 | 0% | 8 | 44.44% |
| 5–7 | Moderate or Moderately Impacted | 8 | 44.44% | 4 | 22.22% |
| 7-9 | Good or Mildly Impacted | 8 | 44.44% | 5 | 27.77% |
| 9–10 | Excellent | 2 | 11.11 | 0 | 0% |

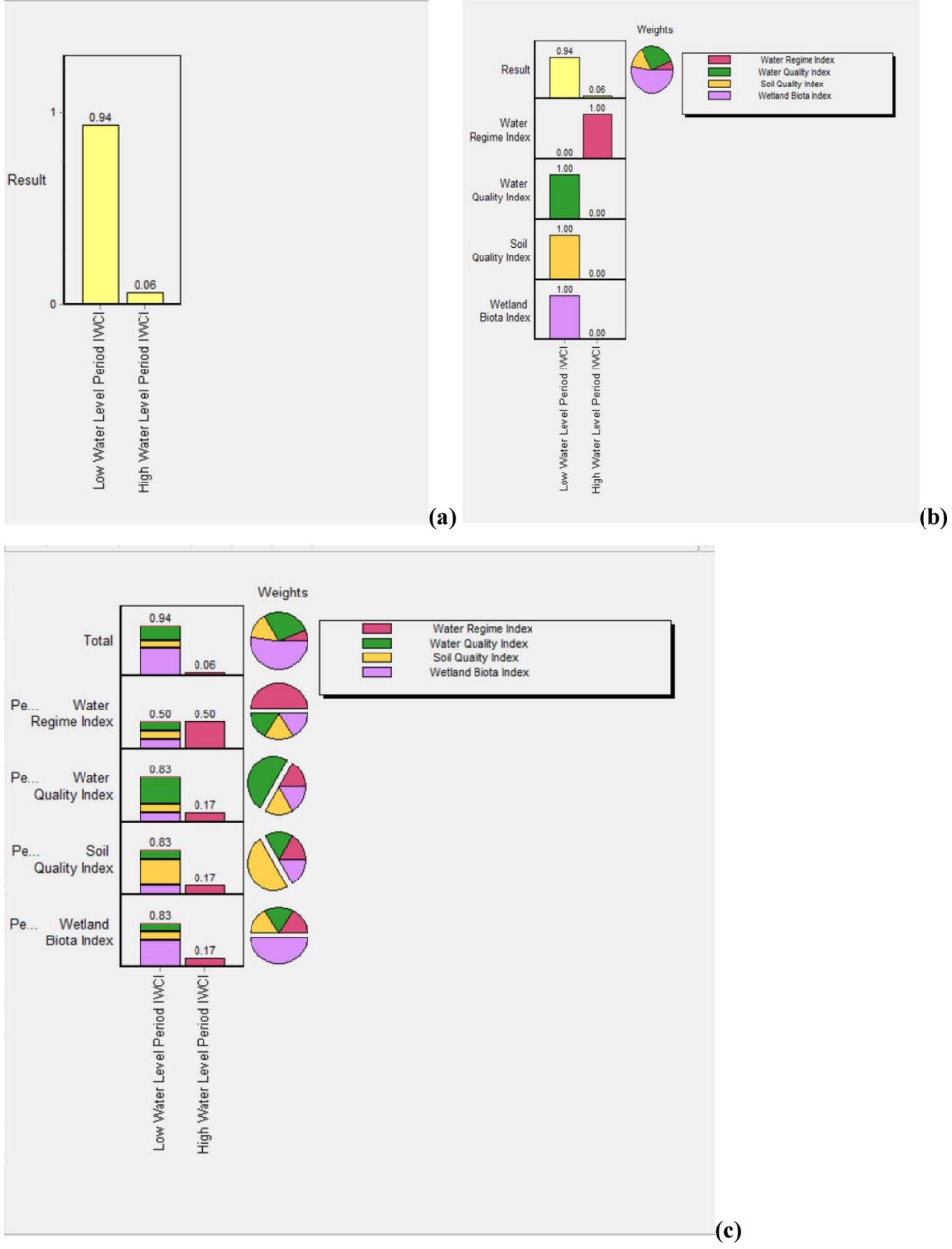

**Fig 2. Ranking of Low water level and high water level periods of IWCI using DEFINITE Software.** *(a) Ranks of Low water level and high water level periods of IWCI using Normal Bar Charts, (b) Contributions of each four sub-indices in each two periods of IWCI, and (c) Total scores of each Low water level and high water level periods of IWCI using various perspectives on group weights.*

level period (0.94), better wetland condition reflected than high water level period which scores 0.06. Ranks of alternative IWCIs (Fig 2a) showed that Low water level period IWCI scored highest from all sub-indices except Water Regime Index (WRI). In the case of various perspectives on group weights, rankings are linked to perspectives, each emphasizing a particular interest in the decision. Perspectives are linked to groups of effects. However, high water level period is highly preferred to monitor the lacustrine wetland condition health in terms of Water Regime Index (WRI) such as: (a) water

depths in meters; (b) extent of water cover in hectares (ha); (c) durations of inundation in days; and (d) rates of recession or refill in m/year were included in this study (Fig 2c).

The two seasons that defined the wetland hydrology phases, namely the pre-rainy season (low water level) of May 2021 and the post-rainy season (high water level) of October 2021 greatly influenced the scores of four sub-indices that included in the computation of IWCI of the LFWs in Lake Tana. As a result of this, the overall integrated wetland condition index of LFWs in Lake Tana from the two season or contrasting hydrological period were very different. IWCI of LFWs in Lake Tana were sensitive to seasonal changes in environmental factors and its effect on lacustrine wetland biotas. The major environmental factors, which changed between low and high water level periods were hydrological regimes (extent of water cover or inundated area (%). As the water level of the Lake Tana increases, lacustrine wetlands receive water from the lake and extent of water cover within the wetland or inundated area (%) also increases. Thus, high water level of the lacustrine wetlands reduced access to recession agriculture and livestock free grazing to the wetland habitats. During the rainy season and post rainy season from the month of July up to November animals shift from Yiganda and Gadero wetlands to the upper watershed, non-flooded alternative grazing areas that serve only for rainy season, thereby allowing Yiganda and Gadero wetlands to support much larger livestock populations than would otherwise be possible. These changes were reflected in higher scores for the post-rainy season or high water level period for the sub-indices of WRI (wetland inundated area in percentage and rate of refill in meter/year).Besides hydrological regimes, wetland biota index(macrophyte index) were changed between the two seasons. The lower score assigned for high water level period for the wetland biota index (macrophyte index) resulting from the widespread of invasive exotic species of *Eichhornia crassipes.*

## Discussion

### Wetland water chemistry conditions in comparison with past data

In comparison with past data of water chemistry, the observed pH values (pH = 6.58–8.64) during low water level and (pH = 7.40–9.98) during high water level almost fell within the pH range (pH = 7.6–8.9) reported by [50]. The range value of the water temperature (18.20–32.10°$_C$) has a wider range in this study as compared to the range value (24.15–26.34°$_C$) reported by [50].The mean EC values of 133.6 µS/cm and 138 µS/cm during low and high water levels, respectively, of this study fell within the EC range (152.48–164.18 µS/cm) and (111–305 µS/cm) reported by [50,51], respectively. The values of total phosphorus in water (TP = 0.23–2.39 mg/l) during low water level and (TP = 2.02–3.14 mg/l) during high water level of this study are higher than those reported by [50] total phosphorus in water (TP = 0.22–1.00 mg/l) and [52] total phosphorus in water (TP = 0.44–1.73 mg/l). The absence of well-established buffer zones and nutrient enrichment of the lake's littoral zone from human sources in the catchment area are likely to be responsible for the rising trend in total nitrogen and total phosphorus in the water. The rising trend in nutrient levels is closely linked to runoff from agriculture and sewage from homes and businesses that is dumped into lakes [50,53].

### Wetland biota assemblages in comparison with past data

A total of 14 macrophyte species were identified in this study, twice as small as the macrophyte species (30) identified by [50]. Almost all macrophyte species reported both during low and high water level phases were listed in the study of [50,54]. The macrophyte species diversity of shore area wetlands in Lake Tana is low compared with the previous works of [50].This could be related to the expansion of free-floating species invasive species of Eichhornia crassipes (water hyacinth).The total number of zooplankton taxa in our study is smaller than a total of 57 zooplankton taxa representing Cladocerans (24 taxa), Copepods (9 species), and Rotifers (24 taxa) by [55].However, zooplankton taxa richness in our wetlands (31 zooplankton taxa) was higher than the review work of [56] who reviewed that several zooplankton studies in Lake Tana found a total of 26 zooplankton species, including 3 copepods, 11 cladocerans, and 12 rotifers. The total

number of zooplankton taxa reported in the current study was also higher; a total of 13 species, four copepods, and nine cladocerans were identified by [57].

**Present integrated wetland condition index in relation to the previous condition index**

The index of lacustrine wetland condition was developed based on four wetland characteristics tested to evaluate the quality of selected lacustrine wetlands around Lake Tana (hydrology, wetland water physiochemical environment, wetland soil environment, and biota). This index design method could be extrapolated to other systems and thus aid in tracking the restoration and degradation of lacustrine wetlands. These four wetland characteristics applied for ILWCI development were similar to those used by [2], who developed a wetland condition index by integrating catchments, physical form, soils, water chemistry, hydrology, and biota for Victoria, Australia, except catchments and physical form, which were not integrated in this study. Our integrated wetland condition indices also agreed with [18] who developed standardized measures of coastal wetland conditions for the Laurentian Great Lakes Basin-Wide Scale by integrating physico-chemical conditions with wetland biota. From the wetland biota category, birds, anurans, and fish that were considered for the Laurentian Great Lakes Basin-Wide Scale were not included in this study. [18] did not integrate hydrology with coastal wetland conditions for the Laurentian Great Lakes Basin-Wide Scale to develop wetland condition indices. [19] developed an integrated wetland assessment tool for condition and health (WATCH) by incorporating horizontal position, vertical position, biology, hydrology, soil condition, and water chemistry similar to this study, except horizontal position and vertical position, which were not included in this work (Table 5).

When we compared the sub-indices included in the development of the current IWCI with the previous IWCI developed by different authors, the sub-indices of the horizontal position, vertical position, landscape development intensity index (LDI), birds, anurans, fish, and wetland buffer considered by previous authors were not included in this IWCI development. Method of sub-indices scoring and overall WCI development of this study similar with [12] except ISA using PCORD software.

[12] compared water and soil parameters among wetlands within three a priori land use categories (reference, agricultural, and urban) using Fisher's LSD pairwise comparison using Minitab Statistical Software (2000) for FWCI development (Table 5). Similarly in this study, hydrology, water, and soil parameters within three wetland impact level categories were compared using Friedman post-hoc, Fisher's LSD pair-wise comparison in r.

Six macrophyte metrics were selected for inclusion in the development of the Florida Wetland Condition Index (FWCI) for depressional forested wetlands, including tolerant and sensitive indicator species; the Floristic Quality Assessment Index (FQAI); exotic species; native perennial species; and wetland status species [12]. Likewise, for the development of IWCI for LFWs in Lake Tana, three macrophyte metrics, such as tolerant indicator species, sensitive indicator species, exotic species, and wetland status species, were selected. Besides three macrophyte metrics, Tolerant Diatom Indicator Species, Tolerant Diatom Indicator Species, and pH Class 3 (circumneutral, mainly occurring at pH values around 7) were selected for Diatom metrics, while sensitive and tolerant zooplankton indicator species were included for Zooplankton metrics. However, both Diatom and zooplankton indicator species were not considered for FWCI of depressional forested wetlands.

In this research, the lists of indicator species were determined using indicator species analysis (IndVal and p-value tests) using species-level abundance data using PAST software [41].To determine the characteristic species of each habitat, the IndVal technique combines metrics of habitat fidelity (frequency within that habitat type) and specificity (uniqueness to a single site). Previous studies also applied indicator species analysis (IndVal and p value tests) for different wetland indicator species. [58] conducted indicator species analysis with the indicspecies package using R to determine the species of the three main zooplankton groups (rotifers, cladocerans, and copepods) that correspond to various trophic stages or are useful indicators in the reservoirs in Spain's Ebro watershed. [12] analyzed indicator species analysis (ISA) using PCORD software (1999) for the development of FWCI of depressional forested wetlands. Furthermore, an indicator

**Table 5. Comparison of the present IWCI with some previously developed wetland condition index across their locations, metrics/sub-indices methods of ISA and overall WCI development.**

| Authors | Location | Metrics/Sub-indices | Method of sub-indices and overall WCI development |
|---|---|---|---|
| **Present Study** | LFWs in Lake Tana, Ethiopia | Hydrological Regime, Water Quality, Sediment Quality & Wetland Biota Index(Macrophyte, Diatom & Zooplankton) | Fisher's LSD pair wise comparison of hydrology, water and soil parameters conducted using R software. PAST software was used to conduct SIMPER analysis, ISA, and ANOSIM testing. The overall WCI was calculated by multiplying each sub-index score by its respective final weight and summing the scores. The final WCI was classified into 5 categories, namely: very poor (0–3), poor (3–5), moderate (5–7), good (7–9), and reference condition (9–10). |
| [19] | Delaware Estuary, | Horizontal position, vertical position, biology, hydrology, soil condition, and water chemistry | Criteria and Trajectory Metrics were applied. The developed Wetland Assessment Tool for Condition and Health (WATCH) which could evaluate data against user-defined criteria and trajectories to identify evidence of current and/or future deficiencies. |
| [17] | Humid regions of China. | Developed WCI by combining the Landscape Development Intensity Index (LDI) and the Water Environment Index (WEI) | LDI and WEI were determined by subtracting the LDI and WEI from 10, considering that the LDI and WEI were inversely proportional to the wetland conditions. The restriction and compensation thresholds were used to identify wetlands in poor and good conditions. |
| [18] | Laurentian Great Lakes Basin | Integrating birds, anurans, fish, macroinvertebrates, vegetation, and physico-chemical conditions | Qualitative evaluations within proportionate ranges were assigned to the total indicator scores. Based on the following criteria, scores were assigned one of five qualitative ratings: extremely poor quality (0–10), low quality (11–20), medium quality (21–30), moderately high quality (31–40), and high quality (41–50). |
| [7] | Nanticoke watershed, USA | Integrating, vegetation, hydrology, and buffer. | All variables that passed the responsiveness, redundancy, and range check were averaged to determine the IWC. Weights of the individual variables were adjusted to reflect wetland ecology and to include variables that represented the vegetation, hydrology, and buffer of a wetland. High, medium, and low site disturbance classes were distinguished by the final IWC score. |
| [16] | Victoria, Australia | Integrating catchments, physical form, soils, water chemistry, hydrology, and biota | The final IWC total score were calculated by multiplying each sub-index score by its respective final weight and summing the scores. Each sub-index had a maximum score of 20. After the weights were applied, the maximum possible total score was 38.4, which, has been scaled to 10 by dividing the total score by 38.4 and multiplying by 10. The final WCI was classified into 5 categories, namely: very poor, poor, moderate, good, and reference condition. |
| [12] | Isolated depressional forested wetlands in Florida, USA | Integrating Water and soil parameters with macrophyte community composition | Fisher's LSD was used to compare soil and water parameters pairwise with ISA using PCORD software (1999). On a scale of 0–10, the reference wetland condition was represented by 10. The FWCI was created by adding the metric scores for each sample wetland. |
| [14] | South-eastern Australia | Measurement and the scoring of indicator value was based on four attributes of wetlands: soil, fringing vegetation, aquatic vegetation and water quality | Measurement and the scoring of indicator values by allocating scores based published information and the authors' own observations. Scores were allocated on a scale from 0 to 4. |

species analysis was conducted by [59] to define juvenile fish indicator species responsible for the differences between the groups identified through the RDA.

The comparison results of water and soil parameters among wetlands within three categories (LI, HI, and RMI groups) using Fisher's LSD pairwise comparison obtained in this study were also compared to work by [12]. [12] comparison showed that: (a) reference wetlands had significantly different water column turbidity, pH, and total phosphorus (TP) concentration than agricultural and urban wetlands; (b) specific conductivity was significantly different between reference and urban wetlands; (c) soil TP was significantly different between reference and agricultural wetlands; and (d) soil organic matter was significantly different between agricultural and urban wetlands. The comparison results of water and soil

parameters among wetlands within three categories (LI, HI, and RMI groups) of this study showed a significant difference (p < 0.05) among less impacted wetlands, river mouth-influenced wetlands, and highly impacted in three hydrological regimes (inundated, exposed and rates of recession), three water parameters (water pH, EC, and TDS) and two sediment parameters (sediment depth and SOM).

Variations in water and soil parameters between less impacted and river mouth influenced wetlands of this study were greater in low water level period than high water level. During high-water phase, the spatial variations were much smaller for most of the measured limnological variables except exposed area (x = 18.72 ± 2.98) and p = 0.0000***, SOM (x = 4.77 ± 0.72) and p = 0.0100** than river mouth-influenced wetlands less wetlands of SOM (X = 2.81 ± 1.34. During high water level period, lacustrine wetlands, river mouth influenced wetlands, the Lake Tana, and its floodplains itself are reconnected. High water levels in hydrological connectivity facilitated the exchange of water, sediments, and organisms among lacustrine wetlands, river mouth-influenced wetlands, Lake Tana, and its floodplains, leading to low spatial environmental heterogeneity and biotic homogenization in the lacustrine wetlands landscape. It backed up the hypothesis that increase water level leads to low spatial environmental heterogeneity and biotic homogenization in the flood plain wetland landscape [60,61].

Results from the few published studies on wetland macrophyte, diatom and zooplankton indicator species identification were consistent with our findings. Only two invasive exotic species observed in the present study during low water level period were *Azolla fern* and *Eichhornia crassipes.* [45] also identified Eichhornia crassipes as macrophyte exotic species in Isolated Depressional Forested Wetlands, Florida.

Sensitive diatom indicator species of the less impacted wetland sites observed in the present study during low water level period were *Aulacoseira granulata*, *Cymbella sp.*, *Diploneis parma*, *Epithemia argus, Navicula placenta, Navicula pseudotuscula, Nitzschia dissipata*, *Pinnularia sp.*, *Surirella linearis*, *and Synedra sp.* Cymbella sp., Navicula sp., and *Nitzschia sp.,* were also selected as diatom sensitive indicator species for the development of FWCI for Isolated Depressional Forested Wetlands. Tolerant diatom indicator species observed in highly impacted wetlands of the present study during low water level period *(Navicula sp.* and *Pleurosigma angulatum)* were different from species identified by [45], namely: *Cyclotella pseudostelliger, Diploneis elliptica, Navicula confervacea, Navicula minima, Navicula mutica, Navicula recens, Navicula subminuscula, Neidium alpinum, Nitzschia subacicularis, Pinnularia braunii, Pinnularia divergentissima* and *Stauroneis kriegeri.*

The five tolerant zooplankton indicator species observed in the present study at highly impacted wetlands with the highest indicator values included *Bosmina longirostris*, *Brachionus patulus*, *Brachionus sp.*, *Keratella tropica*, and *Mesocyclops*. Additionally, [8] reported that degraded wetlands were dominated by species that could withstand pollution, such as *Brachionus* also found pollution-tolerant taxa (e.g., Brachionus) dominated degraded wetlands. During the high water level period in the present study, plant-associated Cladocerans (*Ceriodaphnia sp.*), and Cyclopoid copepods (*Thermocyclops*), were identified as significant sensitive zooplankton indicator species of the less impacted wetland sites.

Although this IWCI developed and tested in this study could be a tool to inform decision-makers on lacustrine wetland conservation and restoration priorities throughout the Lake Tana Ecosystem, there are some limitations of the methods used in the study as well as the proposed index and future prospects of the study that needed to be strengthened. Factors that limit the development and test of IWCI for lacustrine fringe wetlands in Lake Tana were (a) limitations in spatial scale (only six selected wetlands were not sufficiently representative of the entire lacustrine wetland in Lake Tana). (b) limitation in temporal scale (the study was conducted only during two seasons: the pre-rainy (drawdown phase) of May 2021 and the post-rainy (flooding phase) of October 2021, i.e., the rest of the dry season of water level declining (recession phase) and the rainy season of water level rising (refill phase) were not covered). (c) IWCI does not sufficiently discriminate reference from impaired sites (CCP value of below 70%). Low CCP value could be associated with the limitation of the prior knowledge classification approach. The precise reference sites could not be identified since the selection of sites was based on a prior understanding of the effect pressures caused by human disturbances. This could influence the overall index value. Therefore, reference and impacted sites need to be chosen using a post-classification approach rather than

prior knowledge of the impact pressures caused by human disturbances for future research projects on the development of the wetland state index. (d) Indicators at the landscape level, wetland biota of fish, and wetland birds as indicator species were not included in this research work. Thus, further research needs to be done by including landscape, fish, birds, and long-term water level variability indicators in the future.

## Conclusion

In this study, an integrated wetland condition index (IWCI) developed and tested for lacustrine fringe wetlands (LFWs) in Lake Tana appears holistic and adequately provides sufficient information about the lacustrine wetland condition for sustainable management and restoration options of lacustrine wetlands. According to the findings of this study, the overall integrated wetland condition index of LFWs in Lake Tana from the two-season or contrasting hydrological period was very different. IWCI of LFWs in Lake Tana were sensitive to seasonal changes in environmental factors and their effect on lacustrine wetland biotas. The major environmental factors that changed between low and high water level periods were hydrological regimes, particularly the extent of water cover or inundated area. The precise reference sites could not be identified since the selection of sites was based on a prior understanding of the effect pressures caused by human disturbances. Therefore, reference and impacted sites needs to be chosen using a post-classification approach rather than a prior knowledge of the impact pressures caused by human disturbances for future research projects on the development of the wetland state index.

Our findings could contribute to informing decision-makers on lacustrine wetland conservation and restoration priorities around Lake Tana, developing rehabilitation plans, and monitoring and judging any future progress in the conservation and restoration conditions of lacustrine wetlands in Lake Tana. In order to follow the restoration and degradation of lacustrine wetlands, this integrated wetland condition index method may be extrapolated to different systems.

Indicators at landscape level, wetland biota of fish, and wetland birds as indicator species were not included in this research work. Besides, long-term water level variability (the deviation of monthly mean water levels over 30 years of records for Lake Tana) is an important indicator of potential impacts on lacustrine ecosystems. However, long-term water level variability was not considered in this work. Thus, further research needs to be done by including landscape, fish, birds, and long-term water level variability indicators in the future. Establishing lacustrine fringe wetlands monitoring sites and keeping an eye on their state using IWCI created and tested by this study are necessary for maintaining and restoring lacustrine fringe wetlands as well as formulating policy.

## Supporting information

**S1 Table. Score allocation for selected wetland hydrology, water chemistry, and sediment characteristics of LFWs in Lake Tana.**
(TIF)

**S2 Table. Indicator species analysis (ISA) results of macrophyte species with their IndVal (%) and p value.** *Bold values indicate significant P values at* $p < 0.10$.
(TIF)

**S3 Table. SIMPER results listing the macrophyte species that contributed to the average Bray-Curtis dissimilarity between all pair of sites among the three impact categories of wetlands.** *Shaded areas indicated macrophyte taxa that did not identified in the given season.*
(TIF)

**S4 Table. Indicator species analysis (ISA) results of diatom species with their IndVal (%) and p value during pre-rainy season.** *Bold values indicate significant at* $p < 0.10$.
(TIF)

**S5 Table.** Indicator species analysis (ISA) results of diatom species with their IndVal (%) and p value during post-rainy season.
(TIF)

**S6 Table. SIMPER results listing the diatom taxa that contributed to the average Bray-Curtis dissimilarity between all pair of sites among the three impact categories of wetlands.** *Open areas indicated diatom species that did not identified in the given season.*
(TIF)

**S7 Table. Indicator species analysis (ISA) results of zooplankton taxa with their IndVal (%) and p value.** *Bold values indicate significant P values at* $p < 0.10$.
(TIF)

**S8 Table. SIMPER results listing the zooplankton taxa that contributed to the average Bray-Curtis dissimilarity between all pair of sites among the three impact categories of wetlands.**
(TIF)

**S9 Table. Summarized scoring values of water regime, water quality and sediment quality sub-indices.**
(TIF)

**S10 Table. Sample diatom species normalize score values between 0 and 10 using [46] equation.**
(TIF)

**S11 Table. Summary of Macrophytes, Diatom, Zooplankton and Overall biota index values.** *Where **values less than 0 are interpreted as 0, and * values more than 10 as 10.*
(TIF)

**S12 Table. Multiplying each sub-index score by its respective final weight and summing the scores of overall wetland condition index.**
(TIF)

**S1 Figure. Macrophyte indicator species test result of three impact categories of wetlands at p < 0.05 boxed: (a) during pre-rainy season, and(b) post-rainy season.**
(TIF)

**S2 Figure. Diatom indicator species test result of three impact categories of wetlands at p < 0.05 boxed: (a) during pre-rainy season, and (b) post-rainy season.**
(TIF)

**S3 Figure. Zooplankton indicator species test result of three impact categories of wetlands at p < 0.05 boxed during post-rainy season.**
(TIF)

## Acknowledgments

The authors wish to thank the anonymous reviewers for their valuable comments and recommendations on the original version of this study.

## Author contributions

**Conceptualization:** Yirga Kebede Wondim, Ayalew Wondie Melese.

**Data curation:** Yirga Kebede Wondim.

**Formal analysis:** Yirga Kebede Wondim.

**Funding acquisition:** Ayalew Wondie Melese.

**Investigation:** Yirga Kebede Wondim.

**Methodology:** Yirga Kebede Wondim, Ayalew Wondie Melese.

**Project administration:** Ayalew Wondie Melese.

**Resources:** Yirga Kebede Wondim, Ayalew Wondie Melese.

**Software:** Yirga Kebede Wondim.

**Supervision:** Ayalew Wondie Melese.

**Validation:** Ayalew Wondie Melese.

**Visualization:** Yirga Kebede Wondim.

**Writing – original draft:** Yirga Kebede Wondim.

**Writing – review & editing:** Ayalew Wondie Melese.

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
