## [Decision Letter · Decision Letter 0]

26 Mar 2025

Dear Dr. Wondim,

Thank you for submitting your manuscript to PLOS ONE. After careful consideration, we feel that it has merit but does not fully meet PLOS ONE’s publication criteria as it currently stands. Therefore, we invite you to submit a revised version of the manuscript that addresses the points raised during the review process.

We look forward to receiving your revised manuscript.

Kind regards,

Said Muhammad

Academic Editor

PLOS ONE

Reviewers' comments:

Reviewer's Responses to Questions

**Comments to the Author**

1. Is the manuscript technically sound, and do the data support the conclusions?

Reviewer #1: Yes

Reviewer #2: Yes

2. Has the statistical analysis been performed appropriately and rigorously?

Reviewer #1: Yes

Reviewer #2: Yes

3. Have the authors made all data underlying the findings in their manuscript fully available?

Reviewer #1: Yes

Reviewer #2: Yes

4. Is the manuscript presented in an intelligible fashion and written in standard English?

Reviewer #1: Yes

Reviewer #2: Yes

Reviewer #1: General Comments: Typo mistakes, grammatical and syntax corrections are required in certain places

Abstract:

The abstract is satisfactory.

Introduction:

The introduction provides sufficient details and context.

Materials and Methods:

The rationale for choosing specific post-hoc tests, such as Fisher’s LSD, should be explained in more detail. Additionally, it would be beneficial to briefly discuss alternative methods, like Dunn’s test, to help readers understand the reasoning behind the choice.

More clarity is needed regarding the specific statistical functions used in each statistical package. This will improve the reproducibility of the analysis by other researchers. It would be helpful to explicitly mention which software and versions were used, and provide clear references to the specific functions applied for each part of the analysis.

While the overall framework for the scoring system is clear, more detail is required on the rationale behind the system, including the criteria for weighting different components. Additionally, further explanation of how the indices were validated would improve the transparency of the method.

How was the accuracy of the Integrated Lake Wetland Condition Index (ILWCI) validated? Providing details on the validation process will enhance the credibility of the index.

To improve transparency and interpretation, it would be helpful to include a data dictionary or a table summarizing the raw values and scoring systems for each parameter. This would allow readers to better understand the scoring process and the data being used.

Results and Discussion:

Figure 1: This figure needs improvement. Enhancing the clarity, labels, and possibly adding more detail will make it more informative and easier to interpret.

Figure 2: The resolution of this figure should be improved to ensure that all details are clear and legible.

The discussion section would benefit from an acknowledgment of the limitations of the methods used in the study as well as the potential limitations of the proposed Index.

Conclusions:

The conclusions are satisfactory.

Supplementary Materials:

All figures in the Supplementary needs enhancement, the figures shoudl be readable and in clear resolution

Reviewer #2: General Comment:

In the manuscript, an integrated wetland condition index (IWCI) system was tried to develop for lacustrine for a wetland in Ethiopia based on the four conditions of the existing wetlands hydrology, chemistry, water quality and biota using some sort of simulation modelling. The research is appealing, but the quality of information, applied methodologies and findings have major flaws, which need to be validated before the final decision. Some of the specific comments are as follows:

- Abstract- please rewrite the abstract with this flow: issue/purpose, methodology applied, major findings, concluding remarks with a main recommendation. In its present form, it’s very segregated and couldn’t get the outcomes.

- Literature presented in the manuscript must be cited from the recently published literature. Still there information regarding the decision making system is missing in the existing literature. Gap analysis needs to be clearly mentioned at the last of the introduction session.

- Is this the only study done of this specific wetland? There is no other published literature mentioned in the present manuscript

- Why only these few parameters are considered to be investigated? There are other important parameters too?

- The sampling procedure needs to be given in details

- Why could be the sources of these nutrients?

- How the information or data is reliable and reflect for the whole year? The sampling is performed once in a year.

- Continue monitoring can provide the overall conditions of the wetland health and this is one of the missing point in the present study.

**Do you want your identity to be public for this peer review?** For information about this choice, including consent withdrawal, please see our Privacy Policy

Reviewer #1: No

Reviewer #2: No

---

## [Author Response · Author response to Decision Letter 1]

14 Apr 2025

Response to Reviewer-1

Comment: Typo mistakes, grammatical and syntax corrections are required in certain places. Response:Thank you for your comments. Typo mistakes, grammatical and syntax errors were corrected. Comment:The rationale for choosing specific post-hoc tests, such as Fisher’s LSD, should be explained in more detail. Additionally, it would be beneficial to briefly discuss alternative methods, like Dunn’s test, to help readers understand the reasoning behind the choice. Response:Fisher’s LSD pairwise comparison method was chosen in this study for the following reasons: (a) Fisher's LSD is essentially a series of individual t tests, in contrast to the Bonferroni, Tukey, Dennett, and Holm methods; instead of computing the pooled SD from just the two groups under comparison, this makes it more difficult to achieve significance; instead, it computes the pooled SD from all the groups, which increases power; (b) Fisher's LSD method for multiple comparisons is used in ANOVA to create confidence intervals for all pairwise differences between factor level means, whereas Dunn's test is the appropriate nonparametric pairwise multiple-comparison procedure when a Kruskal–Wallis test is rejected; and (c) Fisher’s LSD test implemented in the R software using the "agricolae" package, whereas Dunn's test is implemented for Stata in the dunntest command.

Comment:More clarity is needed regarding the specific statistical functions used in each statistical package. This will improve the reproducibility of the analysis by other researchers. It would be helpful to explicitly mention which software and versions were used, and provide clear references to the specific functions applied for each part of the analysis. Response:Software’s of R version 4.1.3 (2022-03-10) and PAST version 4.14 were used. Fisher’s LSD test was performed using the R package "agricolae”. For complete functionality of agricolae package, other packages such as MASS, nlme, klaR, Cluster, and algDesign were installed and loaded. Indicator species tests (IndVal% and p (raw)), ‘Analysis of Similarities’ (ANOSIM), and ‘Similarity Percentages’ (SIMPER) performed in PAST 4.14.

---

## [Decision Letter · Decision Letter 1]

15 May 2025

Development and use of integrated wetland condition index for lacustrine fringe wetlands of Lake Tana, Northwest Ethiopia

PONE-D-24-52278R1

Dear Dr. Wondim,

We’re pleased to inform you that your manuscript has been judged scientifically suitable for publication and will be formally accepted for publication once it meets all outstanding technical requirements.

Kind regards,

Said Muhammad

Academic Editor

PLOS ONE

Additional Editor Comments (optional):

Reviewers' comments:

Reviewer's Responses to Questions

**Comments to the Author**

Reviewer #1: All comments have been addressed

Reviewer #2: All comments have been addressed

2. Is the manuscript technically sound, and do the data support the conclusions?

Reviewer #1: Yes

Reviewer #2: Yes

3. Has the statistical analysis been performed appropriately and rigorously?

Reviewer #1: Yes

Reviewer #2: Yes

4. Have the authors made all data underlying the findings in their manuscript fully available?

Reviewer #1: Yes

Reviewer #2: Yes

5. Is the manuscript presented in an intelligible fashion and written in standard English?

Reviewer #1: Yes

Reviewer #2: Yes

Reviewer #1: (No Response)

Reviewer #2: (No Response)

**Do you want your identity to be public for this peer review?** For information about this choice, including consent withdrawal, please see our Privacy Policy

Reviewer #1: No

Reviewer #2: No

---

## [Editor Report · Acceptance letter]

PONE-D-24-52278R1

PLOS ONE

Dear Dr. Wondim,

I'm pleased to inform you that your manuscript has been deemed suitable for publication in PLOS ONE. Congratulations! Your manuscript is now being handed over to our production team.

Kind regards,

on behalf of

Dr. Said Muhammad

Academic Editor

PLOS ONE